# *In silico* drug repurposing at the cytoplasmic surface of human aquaporin 1

**Aled R. Lloyd*****, Karl Austin-Muttitt, Jonathan G. L. Mullins**

Genome and Structural Bioinformatics Group, Faculty of Medicine, Health and Life Science, Swansea University, Swansea, Wales, United Kingdom

* 360839@swansea.ac.uk

**Data Availability Statement:** All docking results and molecular dynamics files for this study are available from the figshare database. The relevant DOI numbers are 10.6084/m9.figshare.27312210. v1, https://doi.org/10.6084/m9.figshare.27312189.

## Abstract

Aquaporin 1 (AQP1) is a key channel for water transport in peritoneal dialysis. Inhibition of AQP1 could therefore impair water transport during peritoneal dialysis. It is not known whether inhibition of AQP1 occurs unintentionally due to off-target interactions of administered medications. A high-throughput virtual screening study has been performed to investigate the possible binding of licensed medications to the water pore of human AQP1. A complete model of human AQP1 based on its canonical sequence was assembled using I-TASSER and MODELLER. The model was refined via the incorporation of pore water molecules from a high-resolution yeast aquaporin structure. Docking studies were conducted for the cytoplasmic domain of the AQP1 monomer against a library of all compounds listed in the British National Formulary (BNF), using the PLANTS software with the ChemPLP scoring function. The stability of the best docked conformations within the intrinsic water pore was assessed via short 15 nanosecond molecular dynamics (MD) simulations using the GROMACS-on-Colab utility. Of the 1512 compounds tested, 1002 docking results were obtained, and 198 of these conformations occupied a position within the intrinsic water pore. 30 compounds with promising docking scores were assessed by MD. The docked conformations for dopamine, gabapentin, pregabalin, and methyldopa were stable in these short MD studies. For furosemide and pravastatin, the MD trajectory suggested a binding mode different to the docking result. A small set of compounds which could impede water transport through human AQP1 have been identified in this computational screening study.

## Introduction

### Aquaporin 1

Of the 13 human forms of aquaporin (AQP) identified to date, 8 can be found in the kidney (AQP1, AQP2, AQP3, AQP4, AQP5, AQP6, AQP7, and AQP11) [1]. Aquaporin was first identified serendipitously through investigation of a protein contaminating the process of purifying Rhesus proteins from erythrocytes [2]. This protein was found to be highly permeable to water and present in a wide range of plants, microbials, and animal species [2]. The 2003 Nobel prize for chemistry was awarded for this discovery [3].

v1, 10.6084/m9.figshare.27312174.v1, 10.6084/
m9.figshare.27312168.v1, and 10.6084/m9.
figshare.27312147.v1.

**Funding:** The author(s) received no specific
funding for this work.

**Competing interests:** The authors have declared
that no competing interests exist.

Aquaporins are tetrameric proteins. Each protein subunit contains a pore and the whole tetrameric arrangement forms a larger central pore [4]. Aquaporin 1 (AQP1) transports only water through the monomeric pore while single valence cations are transported through the central pore in a process mediated by cGMP [5]. It is a conventional aquaporin, and as the first discovered human form of the protein it has attracted significant research interest. As soon as early structural studies had been completed, the transit of water through these pores and the selectivity for water alone had been investigated [6–8], and increasing the size of the pore by mutation was found to increase the range of molecules transported through the pore [9, 10]. Two particular regions internal to the pore, called "NPA regions", were found to be important for water selectivity [11–13]. If one of the two NPA regions is missing, water permeability is reduced by half, however if both are missing water permeability is unaffected [14]. Absence of this region does not appear to affect AQP expression, intracellular processing and basic structure [14]. Another region within the pore, known as the selectivity filter (SF), is involved in the exclusion of protons through steric factors and polarity caused by aromatic side chains [12]. The structure of yeast AQP1 with water molecules in transit was determined by X-ray crystallography in 2013 [15]. This structure showed a chain of hydrogen bonded water molecules passing through the pore in each monomer of the tetramer. The dipoles of the water molecule chain invert after the NPA regions [16]. The orientation of the side chains of the NPA motifs act as a valve, limiting the flow of water through the pore in simulation studies using bovine AQP1 [17]. These observations on the movement of water molecules have been confirmed by molecular dynamics (MD) studies of catfish AQP1 [18]. Within the nephron, AQP1 is located in the vasa recta, proximal convoluted tubule and the descending limb of the loop of Henle [1].

AQP1 is expressed throughout the body and is implicated in water transport during peritoneal dialysis. It is the so-called "ultrasmall pore" in the 3 pore model of dialysis [19]. There is evidence that water transport through AQP1 is inhibited by mercury and tetraethyl ammonium through binding within the intrinsic water pore of the monomer [20]. Of currently licensed medication, there is only evidence to support AQP1 inhibition by acetazolamide, bumetanide and furosemide [20–23]. Based on this work, the loop diuretics act on the intracellular side of the protein, but their precise site and mode of action is unclear with two suggested binding sites. The binding of furosemide was 15 times more potent than bumetanide [22]. Bumetanide derivatives have attracted attention in the field of oncology as GMP mediated cation transport through the central pore of the tetramer is thought to be involved in metastatic spread of some cancers [24]. These compounds are believed to act by binding to the AQP1 central pore [25]. While the main diuretic action of furosemide is due to inhibition of the sodium/potassium/chloride cotransporter [26], there is also evidence of furosemide interacting with $Na^+/H^+$ exchanger isoform 3 [27] and carbonic anhydrase [28]. Additionally, furosemide is known to have anti-inflammatory and anti-asthmatic effects independently of its diuretic properties [29] suggesting the drug has a multifactorial mode of action.

It has also been postulated that some of the adverse effects of some antiepileptic medication, such as lamotrigine and topiramate could be due to inhibition of AQP1, although the evidence for this is from extrapolation of oncological studies in mice and studies of other aquaporins [30].

Given that the main clinical applications of AQP1 inhibition relate to water transport, we undertook an *in silico* drug repurposing screen to investigate the binding of existing licensed medication to the cytoplasmic side of the AQP1 water pore. This site was selected because it is an experimentally confirmed mode of water transport inhibition [22, 23].

Molecular docking is a computational method developed to identify the best energetic fit of an interaction between a small molecule and a large molecule such as a protein [31]. The speed at which large numbers of docking simulations can be performed makes docking a cheap and

attractive pharmacological research tool, however studies are limited by indiscrimination between active and non-active substances [32]. Following the publication of a protocol for high throughput post-docking molecular dynamics studies to improve discrimination between binding compounds and non-binding compounds [32], we have implemented an approach for a drug repurposing molecular docking screen followed by high throughput molecular dynamics studies [33].

## Materials and methods

### Structural modelling

As portions of the cytoplasmic chains were incomplete in crystal structures and in order to create a model based solely on human sequences, threading modelling using the I-TASSER server and suite [34–36] was performed using the canonical FASTA sequence for AQP1 on the Uni-Prot database(entry P29972-1) [37]. Structural visualisation and analysis were performed using UCSF Chimera software versions 1.16 and X1.2.5 [38]. Oligomerisation of the monomeric structure obtained from I-TASSER was performed by homology modelling using MOD-ELLER [39].

### Modelling of water in an aquaporin channel

We generated a model of human AQP1 with water molecules within the water pore using the position of water molecules from yeast aquaporin (PDB:3ZOJ) [40] to ensure that protein side chain orientation was compatible with water transport for the docking studies. Fifteen water molecules in the 3ZOJ [15] structure were identified for the modelling of pore water and quaternary structure. The ParMATT structural alignment algorithm was used to align 3ZOJ and our AQP1 model [40]. Protein residues within 4 Angstroms of these 15 water molecules were identified from the 3ZOJ structure and maintained. MODELLER [39] was used to generate a structure based on the I-TASSER structure of AQP1 and the restricted model of 3ZOJ.

### Membrane site identification

The membrane builder function of CHARM-GUI was used for building a lipid bilayer around the assembled tetramer [41–45].

### Docking studies

**Library of compounds.** 3D structures of all 1512 medications listed in the British National Formulary were sought from the NCBI PubChem databases [46]. Once duplicate records and substances for which there was no structure on the PubChem database were excluded, 1002 medicinal compounds were left in the screening library. Where a compound was listed as a mixture; all components of the mixture were sought from the database. The most common reason a structure could not be found was due to the agent's large molecular size with medicines such as vaccines, monoclonal antibodies and haematological products featuring prominently in this group. The list of compounds included and excluded can be found in the supporting information (S1 Table).

**Active site identification.** The cytoplasmic opening of the intrinsic pore of the AQP1 monomer was identified as the site for investigation as drug binding at this location would impede water transport. Evidence that loop diuretics act on the cytoplasmic surface of AQP1 informed this decision [21, 22]. A wide radius of study (20Å) was selected as no data on precise binding to the cytoplasmic opening is available. The radius of study, highlighted in Fig 1,

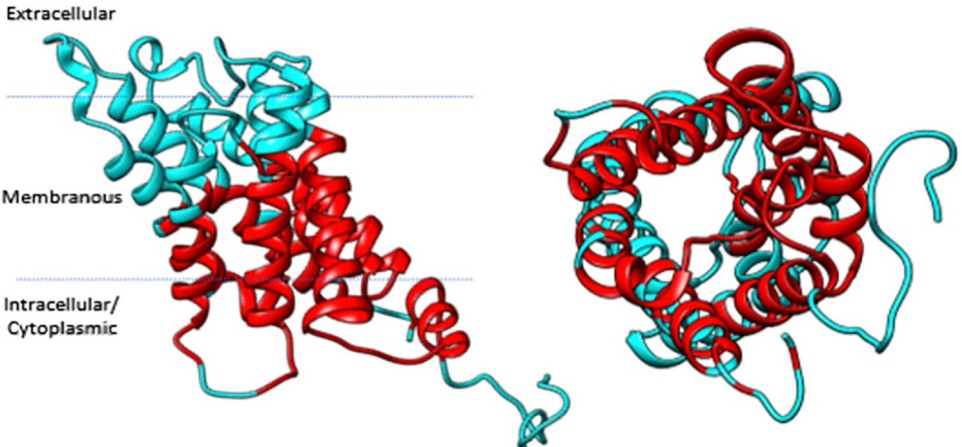

**Fig 1. Human AQP1 monomer with residues within docking search area highlighted in red.** Approximate position of membrane shown by dotted lines.

incorporated the cytoplasmic half of the water pore, the site of mercury inhibition, most of the cytoplasmic chains and membrane regions of the protein.

**Drug binding analysis.** In this work, the docking studies and their analysis have been simplified by adoption of a virtual screening approach, that focuses on only the best docked conformation within a single protein structure. After converting file formats using Open Babel [47], PLANTS software was used to establish the most energetically favourable binding positions of each compound [48, 49]. The binding regions of these structures were classified by visual inspection using UCSF Chimera [38] followed by interrogation of the PLANTS scores for binding energy [49] to establish which compounds were most likely to exhibit a pore blocking interaction with the protein [38]. These processes filtered the docking outcomes to a smaller number for further testing using MD.

## MD studies

We have developed a bespoke cloud based system for preparing MD input files using CHARMM-GUI [41, 42, 44, 45, 50] and running MD simulations using GROMACS [51, 52] entirely on graphics processing units in Google Colaboratory [33].

To prepare the MD system, a membrane bound protein system was assembled using Membrane Builder function of CHARMM-GUI [41–44] and docked ligands prepared using the Ligand Reader and Modeler feature of CHARMM-GUI [53]. The topology and coordinate files were merged and overlapping solvent molecules deleted. We used the TIP3 model for water and POPC to represent the phospholipid bilayer [54]. The energy minimisation and equilibration steps of a GROMACS simulation were then run on a GPU as part of the same system preparation notebook.

15 nanosecond production MD simulations were undertaken to determine which docked ligand poses were the most stable. The CHARMM36 forcefield was used with a pressure of 1 bar and a temperature of 303.15 K. An early stop mechanism was incorporated within the simulation for the point where the root mean square deviation of the ligand exceeds 10Å. Analysis of the MD results was carried out by visual inspection using VMD software [55] and graph analysis of RMSD over time. The RMSD is a median measurement of the displacement between protein and ligand relative to the starting position of the MD simulation. In these simulations the starting position for the ligand is the best conformation in the docking study.

# Results

## Structural data

**Protein structure modelling and validation.**    The canonical FASTA sequence for AQP1 obtained from the UniProt database (P29972-1) was used to obtain I-TASSER structures for AQP1, with a C-score of 1.05 for the best matched model [34, 35, 37]. The initial monomeric and tetrameric structures of AQP1 showed very good visual semblance with published crystal structures for human forms of AQP1. The matchmaker function of USCF Chimera was used to calculate the root mean square deviation (RMSD) of alpha-carbon positions between the experimental model and a mono-meric human AQP1 structure from the Protein Data Bank (PDB 3CSK) [56]. The calculated RMSD between 197 pruned atom pairs was 0.914 angstroms and across all 233 pairs the calculated value was 2.044 angstroms. A refined homology model including pore water molecules located by refer-ence to the yeast AQP1 structure 3ZOJ showed good visual agreement with the backbone of the AQP1 system and the relative positions of water molecules in the channel of the yeast structure. The pore water molecules maintained all relevant hydrogen bonds and sidechain interactions.

## Docking

The most favourable conformations for 1002 drug agents from the BNF yielded 198 compounds bound in or near the opening of the cytoplasmic side of the water pore. These results were further filtered by comparison of docking scores with the docking result for the known binder furosemide. Furthermore, only candidates making 3 or more hydrogen bonds with the protein were considered. A total of 45 compounds were identified as candidates for further investigation via a molecular dynamics approach. The number of compounds selected at each stage is summarised in Fig 2.

Images from the best docked poses of a selection of compounds can be seen in Fig 3. There is variation in binding sites between docked compounds, with gabapentin and dopamine bind-ing deep in the pore and hydrogen bonding to ASN192, while furosemide is predicted to instead interact with several residues at the opening of the pore. The most favourable docking conformations of acetazolamide, lamotrigine and topiramate demonstrated interactions along the cytoplasmic chain between residues PRO233 and LYS267. In our docking results, the most energetically favourable conformation of furosemide binding to the cytoplasmic opening of the intrinsic pore of the AQP1 monomer is different to predictions previously published [22]. The sulphamoyl group of the furosemide molecule binds to a pocket on the cytoplasmic sur-face of the pore opening with the furan ring protruding into the pore. We were unable to reproduce the molecular docking findings of Migliati and co-workers with respect to the bind-ing of bumetanide [22]. The 10 most energetically favourable conformations for the drug iden-tified by PLANTS were all away from the pore.

**Key hydrogen bonds.**   The internal environment of the intrinsic water pore of AQP1 is very hydrophilic with multiple hydrogen donors and acceptors throughout. Several modes of binding have been identified within this virtual screen which encompassed a large proportion of the pore.

Interactions with ASN192 appear to be important for compounds such as gabapentin and dopamine that bind in a position more deeply within the pore. Compounds such as furose-mide that bind closer to the opening of the pore appear to rely on hydrogen bond interactions with ARG93 and THR157.

## Molecular dynamics

A membrane bound AQP1 system was assembled using the membrane builder function of CHARMM-GUI. The position of the membrane in the model was validated by comparison with the annotations in the UniProt database entry for human AQP1, P29972 [37].

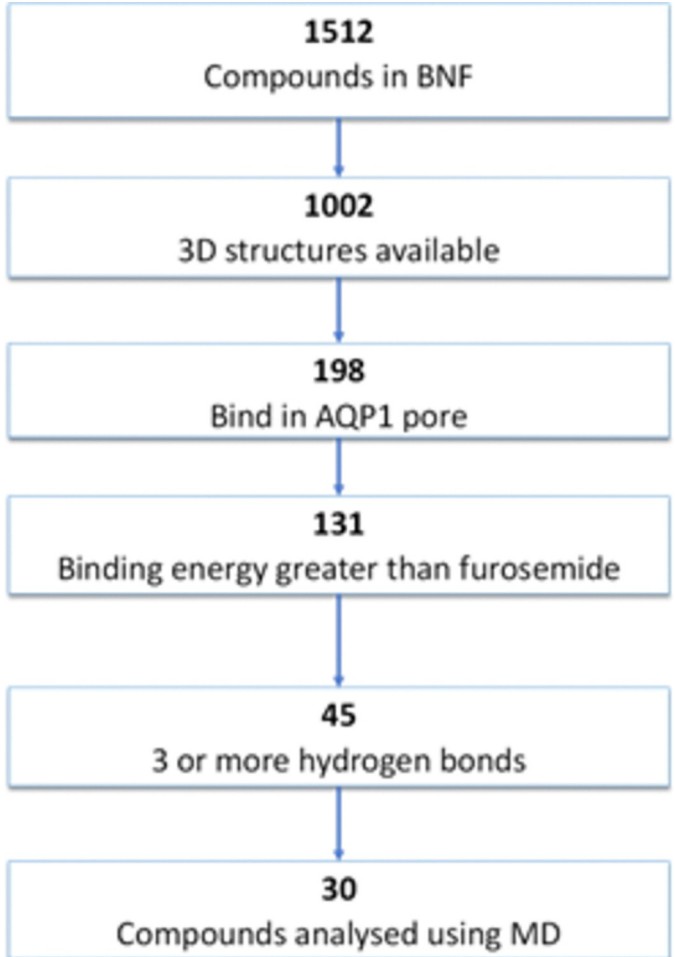

**Fig 2. Flowchart outlining compound selection for molecular dynamic studies from docking study results for all available drug compounds listed in the BNF against the cytoplasmic opening of the water pore of a human AQP1 monomer.**

15 compounds were excluded from the list of 45 either due to similarities to other compounds tested, because of infrequent clinical use or being endogenous compounds. Table 1 summarises the results of the 30 compounds tested with post-docking MD simulations and the excluded compounds are listed in supplementary data (S1 Table).

The duration of MD simulations was an important consideration in the focusing of these studies. Post-docking MD studies of 10, 50 and 100 nanoseconds were assessed by Guterres and Im [32]. The difference in sensitivity measured by the area under a ROC curve (AUC) between each simulation length was small. The calculated AUC values for experiments using the DUD-E dataset were 0.806, 0.840 and 0.836 for simulations of 10, 50 and 100 nanoseconds respectively [32]. Similarly in our study, it was clear that beyond the 15 nanoseconds timeframe, there was consistently no change in the binding status of any of the ligands, i.e. ligands bound at 15ns remained bound and *vice versa*.

Indeed, as can be seen in the graph of Root Mean Square Deviation (RMSD) over time for all compounds in Fig 4, all unstably bound compounds were ejected from the binding site within the first 10 nanoseconds. There are many other recent examples of 10 nanosecond MD studies to assess the stability of docked conformations [57–62].

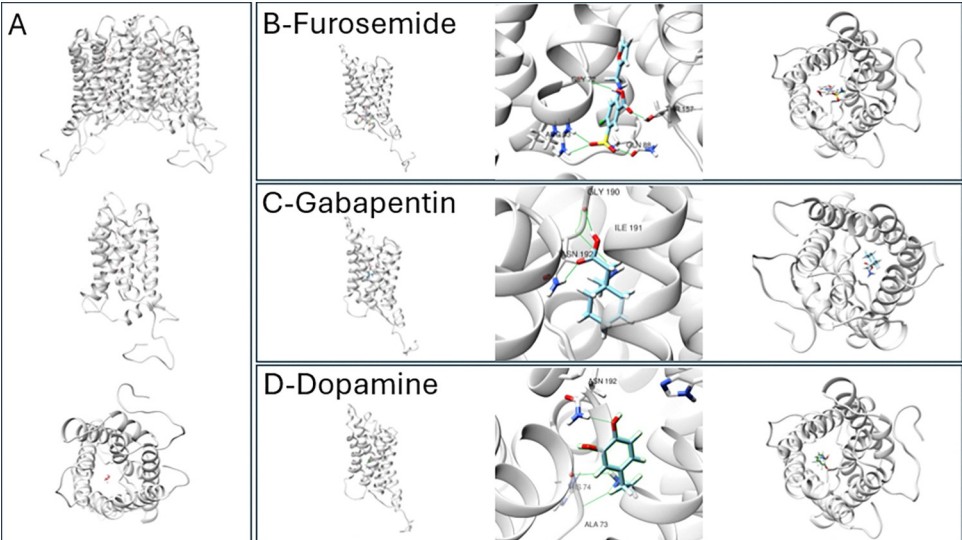

**Fig 3. Panel depicting a selection of the best docked conformations in the water pore of an AQP1 monomer.** A–Images of the human AQP1 model with water molecules in transit, B–the best docked conformation of furosemide, C–the best docked conformation of gabapentin, D–the best docked conformation of Dopamine. Predicted hydrogen bonds are represented by green lines.

Uniquely, furosemide moved from its docked position inside the pore opening, rotating and interacting with the cytoplasmic chain at the opening of the water pore. A conformational change was observed in the cytoplasmic chain from residue PRO233 to the C-terminus (residue LYS 269) on interacting with the furosemide molecule. This phenomenon is illustrated in Fig 5 and was not observed with any other compounds.

The best docked positions for a subset of the indicated compounds which are frequently prescribed–furosemide, levodopa, methyldopa, pravastatin, gabapentin and pregabalin–are highlighted in Fig 3.

**Table 1. Summary of post docking MD studies of drug repurposing screen against the cytoplasmic opening of the water pore of human AQP1.**

| Compound | Final position | Compound | Final position |
|---|---|---|---|
| Methyldopa | Bound | Risedronate | Not bound |
| Dopamine | Bound | Propafenone | Not bound |
| Esmolol | Bound | Tenoxicam | Not bound |
| Panobinostat | Bound | Cytarabine | Not bound |
| Pregabalin | Bound | Carbocisteine | Not bound |
| Gabapentin | Bound | D-Glucose | Not bound |
| Procarbazine | Bound | Safinamide | Not bound |
| Defersariox | Bound | Tranexamic acid | Not bound |
| Glipizide | Bound | Streptozocin | Not bound |
| Furosemide | Bound | Ethambutol | Not bound |
| Famotidine | Bound | Metraminol | Not bound |
| Permetrexed | Bound | Isosorbide dinitrite | Not bound |
| Pravastatin | Bound | Atenolol | Not bound |
| Chloramphenicol | Bound | Noradrenaline | Not bound |
| Fometerol | Bound | | |

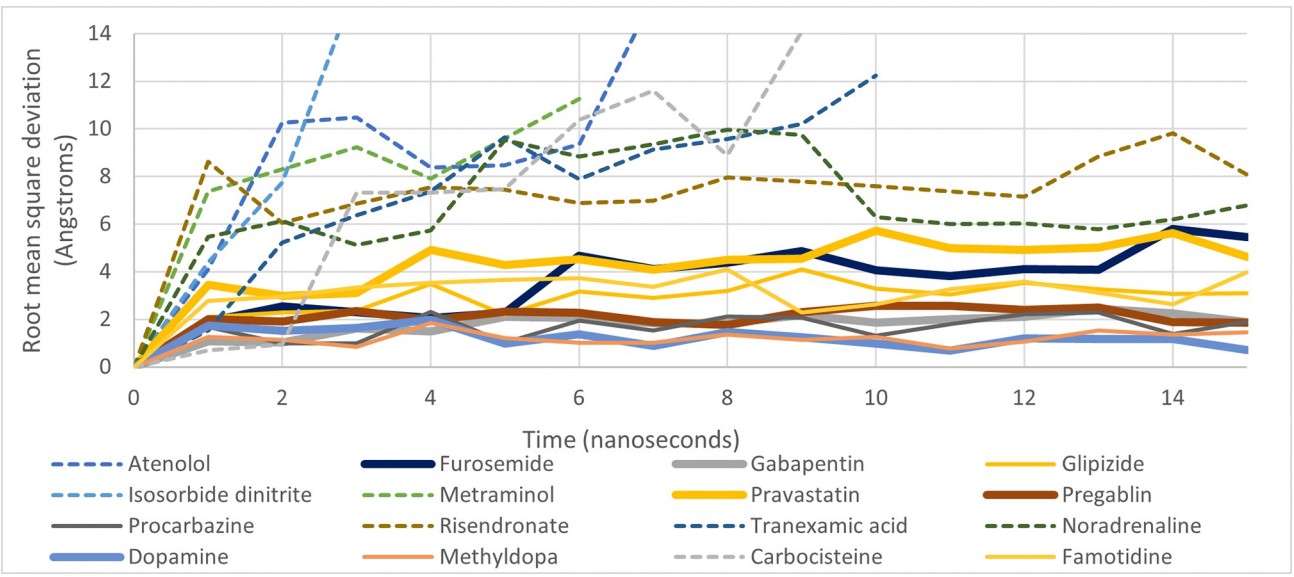

**Fig 4. RMSD over time for a selection of compounds included in the MD simulations.** The compounds assessed as not being bound are indicated with dotted lines, while thick lines indicate bound complexes commonly prescribed.

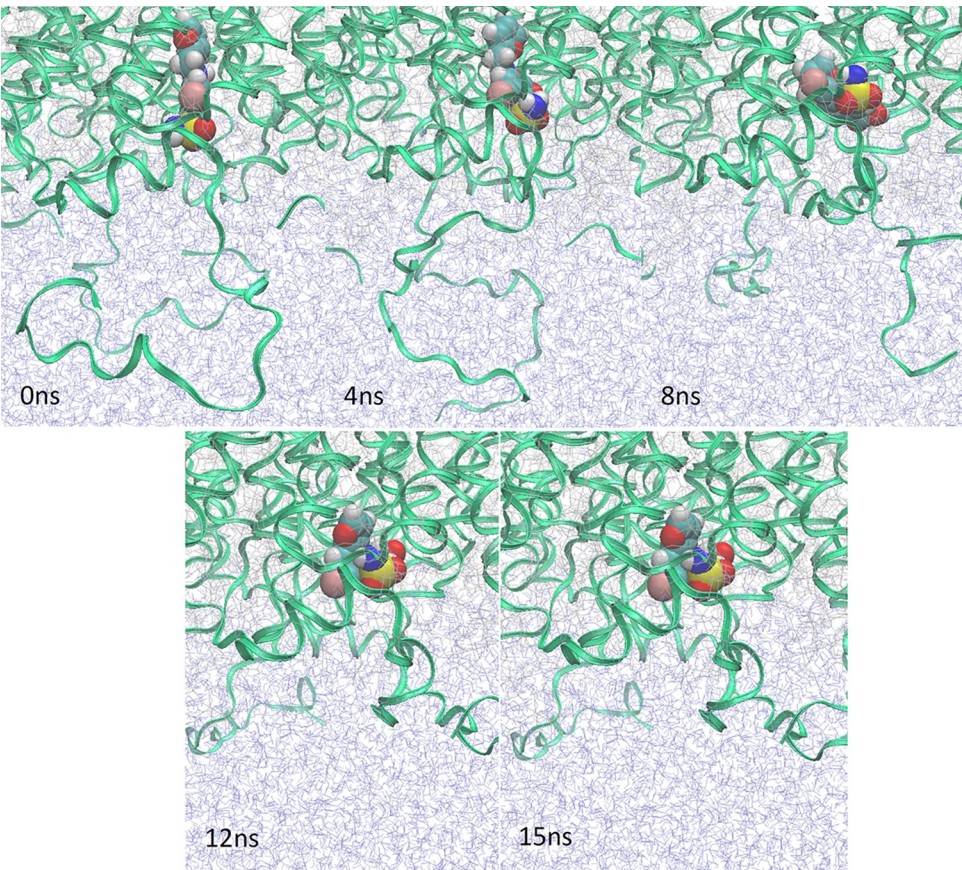

**Fig 5. The position of furosemide and the conformational change in a cytoplasmic domain over the duration of the MD simulation.**

Methyldopa and dopamine both exhibited very low RMSD values during the simulation. The best docking position of both compounds was deep in the pore and at the site of mercury inhibition. Gabapentin and pregabalin exhibited a similar pattern of binding.

The best docking conformation for pravastatin was at the opening of the water pore in a position likely to occlude the flow of water. As the MD simulation progressed, the molecule rotated but otherwise remained over the pore opening.

There was no change as to whether a ligand remained bound in its docked conformation after 10 nanoseconds. Furosemide and pravastatin were both bound to AQP1 in a different position to the starting docked position by this stage.

**Key hydrogen bonds.** The hydrogen bond formed between gabapentin and ASN192 was maintained at the end of the 15 nanosecond MD simulation. The hydrogen bond between ASN192 and dopamine was not maintained with a distance at that stage of 4.08Å between the previous hydrogen bond donor and acceptor atoms. New hydrogen bonds were observed however between the dopamine molecule and residues ALA73 and ASN76. With furosemide, none of the hydrogen bonds in the docked conformation remained by the end of the 15 nanosecond MD simulation, however new hydrogen bonds formed with residues ALA73 and ASN76.

## Discussion

The potential clinical applications of AQP inhibition include treating conditions such as volume overload related to cardiac or renal dysfunction, the treatment of glaucoma or macular oedema and even potentially cerebral oedema due to stroke [20]. As a result there is an emerging effort to identify inhibitors at scale [63]. In contrast to other studies, this work has focussed on existing licensed medication, and aims to investigate the possibility that some adverse effects could be explained by identifying an off-target interaction between the drug and AQP1. There are circumstances where AQP1 inhibition would be undesirable, such as in peritoneal dialysis, where AQP1 activity is thought to play a prominent role [19].

The evidence for AQP1 inhibition by existing licensed medicinal compounds is relatively sparse. Furosemide and bumetanide inhibit water transport through AQP1 by binding to the cytoplasmic side of the protein. Direct binding to the pore and a regulatory action through binding to a cytoplasmic chain have both been proposed as explanations for this effect [21, 22]. In this study, furosemide moved away from a putative docked position at the opening of the pore, eventually settling in a position where a conformational change in the cytoplasmic chain was induced. Based on this simulation, furosemide appears to have several distinct energetically favourable binding positions on the cytoplasmic surface of human AQP1 and may interact with the cytoplasmic chain via a unique mode not observed in any of the other compounds tested.

The choice of search radius in this study was in part influenced by the suggestion that bumetanide could occlude the cytoplasmic opening of the water pore [22], while ensuring that the site of mercury and tetraethylammonium inhibition was also studied [23]. The fact that we were not able to generate an energetically favourable pose for bumetanide within the pore or the cytoplasmic chain of AQP1 reflects its preference in our simulation for hydrophobic interactions. The difference in binding mode compared to furosemide may reflect experimental findings that the binding of bumetanide to AQP1 is 15 times weaker than the binding of furosemide [22]. Bumetanide derivatives have attracted attention in oncology where tubule formation and the migration of cancer cells can be reduced [24, 25]. *In silico* work related to these compounds suggests they bind at the opening of the central ion pore and that their effects are due to the inhibition of cation transport rather than water [25]. The variation in optimal docked poses of the compounds in this polypharmacology screen is interesting. The intrinsic

pore of the AQP1 monomer can accommodate six membered rings, and unsurprisingly for a protein involved in the rapid transfer of a chain of water molecules, provides ample hydrogen bonding opportunities. This accounts for the stable pose of gabapentin in the simulations described here. In general, larger molecules forming multiple hydrogen bonds at the cytoplasmic opening of the water pore were less stable in MD simulations where rotational and entropic factors feature more prominently.

The role of the MD studies in this protocol is to assess the stability of ligand binding and therefore to validate or invalidate results obtained from the docking studies. While root mean square fluctuation and RG values are often used to interrogate ligand-protein interactions in longer simulations, this approach, solely based on RMSD, provides validation data through short MD simulations, facilitating simplified and speedier analysis. However, it is not feasible to make mechanistic assertions relating to the activity of the pore based on these short MD simulations. The stepwise transport of water through the water pore occurs within the time-scale of the simulation but there is no established method to assess whether this process is impeded in the simulations we have performed.

Based on our work it is therefore not possible to characterise the exact nature of water transport inhibition by loop diuretics on AQP1, but possible explanations include a regulatory effect by binding to a cytoplasmic region or direct inhibition via a dynamic process involving multiple sites of interaction.

There is an open question as to whether acetazolamide, lamotrigine, and topiramate affect water transport through AQP1 [23, 30]. If these compounds do affect water transport through AQP1, based on our work, their site of action would be away from the cytoplasmic side of the water pore. The best docked conformations of acetazolamide and lamotrigine identified in this study were in the cytoplasmic region of AQP1. It is interesting that dopamine-like compounds and gabapentinoids are believed to mainly act within the nervous system and it is possible that inhibiting CNS aquaporins may contribute to the observed effects of these drugs.

Interactions between AQP1 and any of methyldopa, levodopa gabapentin, pregabalin or pravastatin are a novel finding. None of these compounds have side effects associated with volume depletion [64], or have ever been investigated for their diuretic properties or for reductions in peritoneal dialysis transport.

AQP1 interacted with a range of different types of compounds in this study, mostly some smaller compounds capable of forming multiple hydrogen bonds, suggesting an element of promiscuity for different ligands. This idea is something that has been studied in kinases [65] and in the case of the cytoplasmic opening of the AQP1 pore is likely the result of an abundance of hydrogen bond donors and acceptors on the protein, and a generous pore diameter near the opening that can accommodate six membered rings.

Corroboration of our findings with experimental work in relation to the cytoplasmic opening of the human AQP1 water pore and investigation of drug binding to the cytoplasmic chains of the protein, particularly the region associated with cGMP binding, are the next steps we recommend following on from this work.

## Conclusion

Traditionally aquaporins have not been considered drug targets, however off-target inhibition of AQP1 could impede peritoneal dialysis efficiency. Loop diuretics are known to interact with the cytoplasmic surface of AQP1 causing inhibition and in this work, this region has been subject to a computational screening study aiming to identify other licensed drug compounds capable of a similar interaction.

The docking studies presented identified 198 compounds bound within the water pore. The most promising conformations underwent a test of stability in the form of a short MD simulation.

Stable *in silico* interactions were observed for several compounds including gabapentin, pregabalin, levodopa and methyldopa. Additionally, possible binding modes were observed for furosemide and pravastatin during their dynamic trajectories, although these were different to their respective docked conformations. Many of these compounds are routinely prescribed to patients receiving peritoneal dialysis, with gabapentin, pregabalin, and furosemide being particularly common.

The results of this virtual screening study indicate the potential for off-target interactions between AQP1 and several drugs commonly administered to peritoneal dialysis patients, which could alter the effectiveness of the dialysis. The identification of these putative interactions presents new experimental targets for *in vitro* and *in vivo* validation.

## Supporting information

**S1 Table. Compounds excluded from MD study and the reason for exclusion.**
(PDF)

## Author Contributions

**Conceptualization:** Aled R. Lloyd, Karl Austin-Muttitt, Jonathan G. L. Mullins.

**Data curation:** Aled R. Lloyd.

**Formal analysis:** Aled R. Lloyd.

**Investigation:** Aled R. Lloyd, Karl Austin-Muttitt.

**Methodology:** Aled R. Lloyd, Karl Austin-Muttitt, Jonathan G. L. Mullins.

**Project administration:** Aled R. Lloyd.

**Resources:** Aled R. Lloyd, Karl Austin-Muttitt.

**Software:** Aled R. Lloyd, Karl Austin-Muttitt.

**Supervision:** Karl Austin-Muttitt, Jonathan G. L. Mullins.

**Validation:** Aled R. Lloyd.

**Visualization:** Aled R. Lloyd.

**Writing – original draft:** Aled R. Lloyd.

**Writing – review & editing:** Aled R. Lloyd, Karl Austin-Muttitt, Jonathan G. L. Mullins.

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
