## [Decision Letter · Decision Letter 0]

8 Sep 2024

PONE-D-24-34926In silico drug repurposing at the cytoplasmic surface of human aquaporin 1PLOS ONE

Dear Dr. Lloyd,

Thank you for submitting your manuscript to PLOS ONE. After careful consideration, we feel that it has merit but does not fully meet PLOS ONE’s publication criteria as it currently stands. Therefore, we invite you to submit a revised version of the manuscript that addresses the points raised during the review process.

We look forward to receiving your revised manuscript.

Kind regards,

Bijay Kumar Behera, Ph.D.

Academic Editor

PLOS ONE

Journal Requirements: When submitting your revision, we need you to address these additional requirements. 1. Please ensure that your manuscript meets PLOS ONE's style requirements, including those for file naming. The PLOS ONE style templates can be found at https://journals.plos.org/plosone/s/file?id=wjVg/PLOSOne_formatting_sample_main_body.pdf and https://journals.plos.org/plosone/s/file?id=ba62/PLOSOne_formatting_sample_title_authors_affiliations.pdf 2. Please note that PLOS ONE has specific guidelines on code sharing for submissions in which author-generated code underpins the findings in the manuscript. In these cases, we expect all author-generated code to be made available without restrictions upon publication of the work. Please review our guidelines at https://journals.plos.org/plosone/s/materials-and-software-sharing#loc-sharing-code and ensure that your code is shared in a way that follows best practice and facilitates reproducibility and reuse. 3. Please ensure that you refer to Figure 1 and 5 in your text as, if accepted, production will need this reference to link the reader to the figure. 4. We notice that your supplementary tables are included in the manuscript file. Please remove them and upload them with the file type 'Supporting Information'. Please ensure that each Supporting Information file has a legend listed in the manuscript after the references list. 

**Additional Editor Comments:**

As per the two reviewers recommendations, I reach my decision to Major Revision of the manuscript.

Reviewers' comments:

Reviewer's Responses to Questions

**Comments to the Author**

1. Is the manuscript technically sound, and do the data support the conclusions?

Reviewer #1: Yes

Reviewer #2: Partly

2. Has the statistical analysis been performed appropriately and rigorously? 

Reviewer #1: N/A

Reviewer #2: Yes

3. Have the authors made all data underlying the findings in their manuscript fully available?

Reviewer #1: Yes

Reviewer #2: Yes

4. Is the manuscript presented in an intelligible fashion and written in standard English?

Reviewer #1: Yes

Reviewer #2: No

5. Review Comments to the Author

Reviewer #1: The manuscript entitled “In silico drug repurposing at the cytoplasmic surface of human aquaporin 1” by Lloyd and co-author is well written. However, the manuscript needs further correction.

1. Abstract needs modification

2. In the introduction section, add (1). About aquaporin, (2). Then Aquaporin 1 (3). Add few latest papers like PMID: 34914911 etc.

3. Describe details in Protein structure modelling part like, where is the sequence ID, retrive from where? etc.

4. 20 ns MD simulation is very less.

5. In the MD simulation, your analysis also RMSF and RG and H-bond

6. Model validation part is missing

7. In Fig. 3, 15 ns is very less to draw a final conclusion.

8. Discuss more in the discussion section of the manuscript.

Reviewer #2: The data presented in the manuscript need to be improved.

Abstract and conclusion section need to be well represented.

The recent relevant paper for this article is not well discussed in the section. Also author advised to follow the required suggestion in attached file

6. PLOS authors have the option to publish the peer review history of their article (what does this mean?). If published, this will include your full peer review and any attached files.

Reviewer #1: **Yes: **Ajaya Kumar Rout

Reviewer #2: No

---

## [Author Response · Author response to Decision Letter 0]

31 Oct 2024

Editor Comments

We thank the editor for their comments and clarification of requirements. In response:

1. The article format has been altered to comply with the journal specifications

2. The code used for MD simulations is freely available through github at https://github.com/bioinfkaustin/gromacs-on-colab. 

3. References to figure 1 and figure 5 can now be found in the text.

4. The supplementary data has been moved to a separate file as requested

Reviewer 1

We thank Reviewer 1, Dr Rout, for his comments. In response:

1. The abstract has been entirely rewritten.

2. The requested additions have been made to the introduction

3. This information can be found in the methods and results section of the revised report.

4. We acknowledge that through reporting only 15 nanoseconds of data, the molecular dynamics (MD) elements of this study include relatively short simulations. This important aspect of study design was not a choice taken lightly. MD studies are a valuable addition to the assessment of docking results and have been validated for this purpose1. A validation study using the DUD-E dataset identified a root mean square deviation (RMSD) threshold of 5.5Å was robust in discerning true active poses in MD simulations of 10, 50 and 100 nanosecond durations. In this study the difference in sensitivity measured by the area under a ROC curve (AUC) between each simulation length was relatively small. The calculated AUC values for experiments using the DUD-E dataset were 0.806, 0.840 and 0.836 for simulations of 10, 50 and 100 nanoseconds respectively(25).

The current study is the first performed using a study protocol for high-throughput docking-MD screening that aims to make such structural bioinformatics studies achievable and accessible to new researchers to the field and researchers in training. This study is not alone in selecting a short duration of MD simulation to evaluate docking results with over 50 examples of studies using 10 nanosecond simulations identified on the NCBI PubMed database at the time of writing2–53.

5. Despite a thorough literature review, we have been unable to identify studies validating the use of root mean square fluctuation (RMSF) and RF in short post-docking MD studies. Without this validation it is unclear whether these additional measurements would meaningfully enhance the study conclusions. As outlined above increasing the complexity of data analysis is contrary to the premise of the protocol under development. In response to the points made, more detail has been included regarding selected observed hydrogen bond patterns before and after the MD simulations which can be found in a dedicated section within the results.

6. A dedicated section has been included with information on the validation of the model. This includes confidence scores obtained from I-TASSER and RMSD values against reference structures.

7. As explained in our response to point 4 (above), with the validation provided and strong support across a raft of literature, we sincerely believe that creditable conclusions can be made from 10 nanosecond simulations in this validated setting1, see point 4 above. We accept that wherever possible, the results of any computational study, but especially those investigating ligand binding, should be validated by laboratory experiments or clinical data due to the inherent limitations of the methods. Both the original and rewritten conclusions make this point plainly.

8. The discussion has been extended by the inclusion of a section discussing the rationale for the methods used.

Reviewer 2

We thank Reviewer 2 for their comments. In response:

1. A robust defence of the methods employed within this study and their rationale can be found above. 

2. The abstract and conclusion sections have been entirely rewritten. Redrafting these sections has taken place without knowledge of the specific concerns of the reviewers and we hope that the new versions are satisfactory. More widely, we have also extended the introduction, including additional references to appropriate studies.

3. We assume that the “recent relevant paper” referred to here is the same investigation of a catfish aquaporin recommended by Reviewer 154. We have referred to this article in the text.

Suggestions within the document

4. The abstract has been completely rewritten given uncertainty about which elements of the abstract required improvement.

5. We have extended the introduction, including additional references to appropriate studies.

6. We again assume that the recent publication referenced here is the mechanistic study of water transport through catfish aquaporin by Behera and colleagues 202254. This study investigating the mechanism of water transport through AQP1 is fundamentally different in its aims and design to the assessment of stability of docked ligand conformations in a screening study. As explained in the response to reviewer 1 the differences between 10 and 100 nanosecond MD simulations for discerning true binding from docking study results are modest1.

7. The conclusion has been rewritten as mentioned in point 3.

References

1. Guterres, H. & Im, W. Improving Protein-Ligand Docking Results with High-Throughput Molecular Dynamics Simulations. J. Chem. Inf. Model. 60, 2189–2198 (2020).

2. Agarwal, S. et al. An integrated computational approach of molecular dynamics simulations, receptor binding studies and pharmacophore mapping analysis in search of potent inhibitors against tuberculosis. J Mol Graph Model 83, 17–32 (2018).

3. Ali, M. A., Nath, A., Jannat, M. & Islam, M. M. Direct Synthesis of Diamides from Dicarboxylic Acids with Amines Using Nb2O5 as a Lewis Acid Catalyst and Molecular Docking Studies as Anticancer Agents. ACS Omega 6, 25002–25009 (2021).

4. Aliko, V. et al. ‘From shadows to shores’-quantitative analysis of CuO nanoparticle-induced apoptosis and DNA damage in fish erythrocytes: A multimodal approach combining experimental, image-based quantification, docking and molecular dynamics. Sci Total Environ 906, 167698 (2024).

5. Aloui, M. et al. QSAR modelling, molecular docking, molecular dynamic and ADMET prediction of pyrrolopyrimidine derivatives as novel Bruton’s tyrosine kinase (BTK) inhibitors. Saudi Pharm J 32, 101911 (2024).

6. Amini, R., Moradi, S., Najafi, R., Mazdeh, M. & Taherkhani, A. BACE1 Inhibition Utilizing Organic Compounds Holds Promise as a Potential Treatment for Alzheimer’s and Parkinson’s Diseases. Oxid Med Cell Longev 2024, 6654606 (2024).

7. Askari, F. S. et al. Digging for the discovery of SARS-CoV-2 nsp12 inhibitors: a pharmacophore-based and molecular dynamics simulation study. Future Virol (2022) doi:10.2217/fvl-2022-0054.

8. Azam, M. A. & Jupudi, S. Insight into the structural requirements of thiophene-3-carbonitriles-based MurF inhibitors by 3D-QSAR, molecular docking and molecular dynamics study. J Recept Signal Transduct Res 37, 522–534 (2017).

9. Azam, M. A. & Thathan, J. Pharmacophore generation, atom-based 3D-QSAR and molecular dynamics simulation analyses of pyridine-3-carboxamide-6-yl-urea analogues as potential gyrase B inhibitors. SAR QSAR Environ Res 28, 275–296 (2017).

10. Bandaru, S. et al. Molecular dynamic simulations reveal suboptimal binding of salbutamol in T164I variant of β2 adrenergic receptor. PLoS One 12, e0186666 (2017).

11. Chaudhuri, A., Bera, A. K., Sarkar, I. & Chakraborty, S. Insights from Analysis of Binding Sites of Human Meprins: Screening of Inhibitors by Molecular Dynamics Simulation Study. Comb Chem High Throughput Screen 19, 246–258 (2016).

12. Chauhan, D. et al. Design, synthesis, biological evaluation, and molecular modeling studies of rhodanine derivatives as pancreatic lipase inhibitors. Arch Pharm (Weinheim) 352, e1900029 (2019).

13. Chen, P.-Y. & Han, L.-T. Study on the molecular mechanism of anti-liver cancer effect of Evodiae fructus by network pharmacology and QSAR model. Front Chem 10, 1060500 (2022).

14. da Silva, G. D. et al. In vitro and in silico studies of the larvicidal and anticholinesterase activities of berberine and piperine alkaloids on Rhipicephalus microplus. Ticks Tick Borne Dis 12, 101643 (2021).

15. Das, B. K., Pv, P. & Chakraborty, D. Computational insights into factor affecting the potency of diaryl sulfone analogs as Escherichia coli dihydropteroate synthase inhibitors. Comput Biol Chem 78, 37–52 (2019).

16. Dhameliya, T. M., Nagar, P. R. & Gajjar, N. D. Systematic virtual screening in search of SARS CoV-2 inhibitors against spike glycoprotein: pharmacophore screening, molecular docking, ADMET analysis and MD simulations. Mol Divers 26, 2775–2792 (2022).

17. Gahtori, J., Pant, S. & Srivastava, H. K. Modeling antimalarial and antihuman African trypanosomiasis compounds: a ligand- and structure-based approaches. Mol Divers 24, 1107–1124 (2020).

18. Guerrero-Perilla, C., Bernal, F. A. & Coy-Barrera, E. Insights into the interaction and binding mode of a set of antifungal azoles as inhibitors of potential fungal enzyme-based targets. Mol Divers 22, 929–942 (2018).

19. Halder, S. K. & Elma, F. In silico identification of novel chemical compounds with antituberculosis activity for the inhibition of InhA and EthR proteins from Mycobacterium tuberculosis. J Clin Tuberc Other Mycobact Dis 24, 100246 (2021).

20. Huang, X., Dorhout Mees, E., Vos, P., Hamza, S. & Braam, B. Everything we always wanted to know about furosemide but were afraid to ask. Am J Physiol Renal Physiol 310, F958-971 (2016).

21. Jamal, S., Grover, A. & Grover, S. Machine Learning From Molecular Dynamics Trajectories to Predict Caspase-8 Inhibitors Against Alzheimer’s Disease. Front Pharmacol 10, 780 (2019).

22. Johari, S., Sharma, A., Sinha, S. & Das, A. Integrating pharmacophore mapping, virtual screening, density functional theory, molecular simulation towards the discovery of novel apolipoprotein (apoE ε4) inhibitors. Comput Biol Chem 79, 83–90 (2019).

23. Kamel, E. M. et al. Molecular modeling and DFT studies on the antioxidant activity of Centaurea scoparia flavonoids and molecular dynamics simulation of their interaction with β-lactoglobulin. RSC Adv 13, 12361–12374 (2023).

24. Kamel, E. M. et al. Xanthine Oxidase Inhibitory Activity of Euphorbia peplus L. Phenolics. Comb Chem High Throughput Screen 25, 1336–1344 (2022).

25. Kausar, M. A. et al. Identifying the alpha-glucosidase inhibitory potential of dietary phytochemicals against diabetes mellitus type 2 via molecular interactions and dynamics simulation. Cell Mol Biol (Noisy-le-grand) 67, 16–26 (2022).

26. Khan, A. A., Baildya, N., Dutta, T. & Ghosh, N. N. Inhibitory efficiency of potential drugs against SARS-CoV-2 by blocking human angiotensin converting enzyme-2: Virtual screening and molecular dynamics study. Microb Pathog 152, 104762 (2021).

27. Khan, M. K. A., Akhtar, S. & Arif, J. M. Development of In Silico Protocols to Predict Structural Insights into the Metabolic Activation Pathways of Xenobiotics. Interdiscip Sci 10, 329–345 (2018).

28. Khan, M. F. et al. Dibenzepinones, dibenzoxepines and benzosuberones based p38α MAP kinase inhibitors: Their pharmacophore modelling, 3D-QSAR and docking studies. Comput Biol Med 110, 175–185 (2019).

29. Khedr, M. A., Mohafez, O. M. M. & Al-Haider, I. A. Virtual Screening-Based Discovery of Potent Hypoglycemic Agents: In Silico, Chemical Synthesis and Biological Study. Curr Comput Aided Drug Des 16, 741–756 (2020).

30. Makeneni, S., Thieker, D. F. & Woods, R. J. Applying Pose Clustering and MD Simulations To Eliminate False Positives in Molecular Docking. J. Chem. Inf. Model. 58, 605–614 (2018).

31. Malekipour, M. H., Shirani, F., Moradi, S. & Taherkhani, A. Cinnamic acid derivatives as potential matrix metalloproteinase-9 inhibitors: molecular docking and dynamics simulations. Genomics Inform 21, e9 (2023).

32. Masumi, M. et al. Methicillin-Resistant Staphylococcus aureus: Docking-Based Virtual Screening and Molecular Dynamics Simulations to Identify Potential Penicillin-Binding Protein 2a Inhibitors from Natural Flavonoids. Int J Microbiol 2022, 9130700 (2022).

33. Modi, P., Patel, S. & Chhabria, M. Structure-based design, synthesis and biological evaluation of a newer series of pyrazolo[1,5-a]pyrimidine analogues as potential anti-tubercular agents. Bioorg Chem 87, 240–251 (2019).

34. Moussa, N., Hassan, A. & Gharaghani, S. Pharmacophore model, docking, QSAR, and molecular dynamics simulation studies of substituted cyclic imides and herbal medicines as COX-2 inhibitors. Heliyon 7, e06605 (2021).

35. Muniz Seif, E. J., Icimoto, M. Y. & Silva Júnior, P. I. In silico bioprospecting of receptors associated with the mechanism of action of Rondonin, an antifungal peptide from spider Acanthoscurria rondoniae haemolymph. In Silico Pharmacol 12, 55 (2024).

36. Nagar, P. R., Gajjar, N. D. & Dhameliya, T. M. In search of SARS CoV-2 replication inhibitors: Virtual screening, molecular dynamics simulations and ADMET analysis. J Mol Struct 1246, 131190 (2021).

37. Omoboyowa, D. A. et al. Structure-based discovery of selective CYP17A1 inhibitors for Castration-resistant prostate cancer treatment. Biol Methods Protoc 7, bpab026 (2022).

38. Omoboyowa, D. A., Balogun, T. A., Omomule, O. M. & Saibu, O. A. Identification of Terpenoids From Abrus precatorius Against Parkinson’s Disease Proteins Using In Silico Approach. Bioinform Biol Insights 15, 11779322211050757 (2021).

39. Parcha, P. et al. Identification of natural inhibitors of Bcr-Abl for the treatment of chronic myeloid leukemia. Chem Biol Drug Des 90, 596–608 (2017).

40. Patel, H. M., Shaikh, M., Ahmad, I., Lokwani, D. & Surana, S. J. BREED based de novo hybridization approach: generating novel T790M/C797S-EGFR tyrosine kinase inhibitors to overcome the problem of mutation and resistance in non small cell lung cancer (NSCLC). J Biomol Struct Dyn 39, 2838–2856 (2021).

41. Patel, H. M., Ahmad, I., Pawara, R., Shaikh, M. & Surana, S. In silico search of triple mutant T790M/C797S allosteric inhibitors to conquer acquired resistance problem in non-small cell lung cancer (NSCLC): a combined approach of structure-based virtual screening and molecular dynamics simulation. J Biomol Struct Dyn 39, 1491–1505 (2021).

42. Pola, M., Rajulapati, S. B., Potla Durthi, C., Erva, R. R. & Bhatia, M. In silico modelling and molecular dynamics simulation studies on L-Asparaginase isolated from bacterial endophyte of Ocimum tenuiflorum. Enzyme Microb Technol 117, 32–40 (2018).

43. Rustamzadeh, A. et al. Targeting Caspases 3/6 and Cathepsins L/B May Decrease Laminopathy-Induced Apoptosis in Alzheimer’s Disease. J Alzheimers Dis 101, 211–221 (2024).

44. Shakibay Senobari, Z. et al. Chromone-embedded peptidomimetics and furopyrimidines as highly potent SARS-CoV-2 infection inhibitors: docking and MD simulation study. BMC Res Notes 16, 224 (2023).

45. Shrivastava, A. K., Kumar, S., Sahu, P. S. & Mahapatra, R. K. In silico identification and validation of a novel hypothetical protein in Cryptosporidium hominis and virtual screening of inhibitors as therapeutics. Parasitol Res 116, 1533–1544 (2017).

46. Sridhar, S. N. C., Palawat, S. & Paul, A. T. Design, synthesis, evaluation, and molecular modeling studies of indolyl oxoacetamides as potential pancreatic lipase inhibitors. Arch Pharm (Weinheim) 353, e2000048 (2020).

47. Suman, null, Chaudhary, M. & Nain, V. In silico identification and evaluation of Bacillus subtilis cold shock protein B (cspB)-like plant RNA chaperones. J Biomol Struct Dyn 39, 841–850 (2021).

48. Tania, M. et al. Cordycepin Downregulates Cdk-2 to Interfere with Cell Cycle and Increases Apoptosis by Generating ROS in Cervical Cancer Cells: in vitro and in silico Study. Curr Cancer Drug Targets 19, 152–159 (2019).

49. Tripuraneni, N. S. & Azam, M. A. A combination of pharmacophore modeling, atom-based 3D-QSAR, molecular docking and molecular dynamics simulation studies on PDE4 enzyme inhibitors. J Biomol Struct Dyn 34, 2481–2492 (2016).

50. Uba, A. I. & Yelekçi, K. Carboxylic acid derivatives display potential selectivity for human histone deacetylase 6: Structure-based virtual screening, molecular docking and dynamics simulation studies. Comput Biol Chem 75, 131–142 (2018).

51. Verma, A. K. et al. Identification of 1, 2, 4-Triazine and Its Derivatives Against Lanosterol 14-Demethylase

---

## [Editor Report · Decision Letter 1]

6 Nov 2024

In silico drug repurposing at the cytoplasmic surface of human aquaporin 1

PONE-D-24-34926R1

Dear Dr. Lloyd,

We’re pleased to inform you that your manuscript has been judged scientifically suitable for publication and will be formally accepted for publication once it meets all outstanding technical requirements.

Kind regards,

Bijay Kumar Behera, Ph.D.

Academic Editor

PLOS ONE

Additional Editor Comments (optional):

The authors have revised the reviewers comments. My recommendation is to accept the manuscript for publication.
---

## [Editor Report · Acceptance letter]

8 Nov 2024

PONE-D-24-34926R1 

PLOS ONE

Dear Dr. Lloyd, 

I'm pleased to inform you that your manuscript has been deemed suitable for publication in PLOS ONE. Congratulations! Your manuscript is now being handed over to our production team.

Kind regards, 

on behalf of

Dr. Bijay Kumar Behera 

Academic Editor

PLOS ONE